# Analysis of the Efficiency and Prognostic Value of the Sentinel Node Technique in Oral Squamous Cell Carcinoma after Seven Years

**DOI:** 10.3390/medicina57101092

**Published:** 2021-10-12

**Authors:** Maria Suárez-Ajuria, Abel García-García, José M. Suárez-Peñaranda, Miguel Garrido-Pumar, Cintia M. Chamorro-Petronacci, José M. Somoza-Martín, Mario Pérez-Sayáns

**Affiliations:** 1Oral Medicine and Surgery Unit, Faculty of Dentistry, Universidade de Santiago de Compostela, 15782 Santiago de Compostela, Spain; mariasuaj@gmail.com (M.S.-A.); abel.garcia@usc.es (A.G.-G.); cinolo@gmail.com (J.M.S.-M.); perezsayans@gmail.com (M.P.-S.); 2Instituto de Investigación Sanitaria de Santiago (IDIS), 15782 Santiago de Compostela, Spain; 3Department of Forensic Sciences and Pathology, University of Santiago de Compostela, 15782 Santiago de Compostela, Spain; jm.penaranda@gmail.com; 4Department of Pathology, Clinical University Hospital, 15782 Santiago de Compostela, Spain; 5Department of Nuclear Medicine, Santiago de Compostela’s University Hospital, 15782 Santiago de Compostela, Spain; Miguel.Garrido.Pumar@sergas.es

**Keywords:** sentinel lymph node biopsy, mouth neoplasms, sensitivity, specificity, survival

## Abstract

*Background and objectives*: The purpose of this study was to analyse the diagnostic and prognostic efficiency of the sentinel lymph node biopsy technique (SLNB). *Materials and Methods*: This is a prospective observational study performed by the Hospital Complex in Santiago de Compostela (CHUS) in Spain, between February 2013 and June 2020. The study included 60 patients, who had been diagnosed with OSCC in stage T1/T2N0M0. *Results*: 10 patients (16.7%) presented with SN+ (sentinel node positive). The majority (80%) only presented subcapsular affection, however one case also presented with extracapsular affection. Using the Kaplan–Meier curves, we determined that the average survival estimation for SN− patients was 74.0 months (CI95% 67.6–80.5) and it was 45.4 months (CI95% 10.9–24.0) for SN+ patients (*p* = 0.002). SN+ patients presented an OR = 11.000 (CI95% 2.393–50.589, *p* = 0.002) for cancer-related mortality. In terms of the diagnostic performance of the SN (sentinel node) test, a 55% sensitivity, a 100% specificity, 100% PPV and a 84% NPV were obtained. The analysis using ROC (receiver operating characteristic) curves revealed an AUC = 0.671 (CI95% 0.492–0.850, *p* = 0.046). *Conclusions*: SLNB seems to be an adequate technique for the detection of hidden metastases.

## 1. Introduction

Oral squamous cell carcinoma (OSCC) is the 11th most common cancer in the world [1] and it accounts for 128,000 deaths each year [2]. In the early stage of OSCC (T1-T2/NO) the presence of metastasis lymph nodes reduces the patient’s survival by 50% and it is one of the most important adverse indicators of the disease’s prognosis [3,4,5]. Furthermore, it has been calculated that there is a risk of ganglionic affectation in around 20–30% of the cases [2,6,7,8]; therefore, the correct staging of lymph nodes is crucial for the patient’s therapeutic development. Given that modern image techniques (ultrasound, computed tomography, positron emission tomography and magnetic resonance imaging) [9,10] and physical palpation are not sensitive enough to be able to determine the definitive position of the lymph node, associated surgical techniques must be used [4,11,12,13].

The therapeutic management of OSCC in T1-T2/NO remains a controversial topic. One of the most historically recognised techniques, which is also used in staging, is the lateral neck dissection (3,5,10); a technique that has produced better survival rates than those attained when enforcing the “watch and wait” approach [6]; nonetheless, 70–80% of the patients who are subjected to this technique do not receive any kind of benefit [11,13] given that they do not present with hidden metastatic lymph nodes [14] and, as a result, these patients are exposed to a series of associated postoperative risks, such as shoulder pain and spinal nerve damage [4] that reduce their quality of life [8].

That is why the sentinel lymph node technique (SLNB) has been introduced in OSCC as a much less invasive [12] technique for the adequate staging of the lymph nodes that avoids the postoperative morbidity inherent to lateral dissection [15]. This technique is widely recognised and accepted in the treatment of breast cancer [16] and melanoma [17]; however, in the treatment of head and cervical cancer (HNC), it is only considered as an alternative to lateral neck dissection [4].

The technique shows that, in those patients in which the sentinel nodes (SN), that is to say, the nodes into which the tumour cells will drain from the primary tumour [5] are free of metastasis, it is unlikely that other hidden nodes exist in the lymphatic chain [18]. Therefore, following the local surgical excision of the tumour and the SN, the patient will be completely free of the disease. In the event in which a metastatic affection is present in the lymph nodes, a neck dissection would be required [19]. A considerable number of studies have been published that show a sensitivity and negative predictive values of between 91–95% and 90–98%, respectively [4,15,20]. On the other hand, in the random trial by D’cruz et col which observed a sample of 596 patients, the overall survival after three years in patients who underwent this technique was 80% compared with 67.5% in patients who did not undergo this technique [21].

Lastly, the purpose of this study was to analyse the diagnostic and prognostic efficiency of the SLNB technique in patients with OSCC T1-T2/N0, as well as evaluating the overall survival.

## 2. Materials and Methods

### 2.1. Study Design

This is a prospective observational study, performed by the Maxillo-Facial Surgery Unit of the University Hospital Complex in Santiago de Compostela (CHUS) in Spain. This study was approved by the Autonomous Ethics Committee of Galicia (Spain) under reference 2020/222. This study has been designed according to the STROBE recommendations [22]. The information was gathered between February 2013 and June 2020, which was the last follow-up date. All of the procedures were performed with the understanding and written consent of all of the subjects, in accordance with Helsinki Declaration and its subsequent modifications.

### 2.2. Calculation of Sample Size

According to the hospital’s records, there is an average of 10 stage T1 and T2 tumours that could be treated using the SLNB each year and there is an average of 200 stage 3 and 4 tumours each year. For a 7-year study, with a sample of 60 patients, and based on the comparison of proportions for a population of 1000 patients, with an expected proportion of 10.0% and a confidence level of 95.0%, the precision would be 5.578. This sample size has been calculated using Epidat 4.2 (SERGAS, Galicia, Spain).

### 2.3. Study Population, Inclusion, and Exclusion Criteria

The study included 60 subjects who were recruited prospectively. The inclusion criteria were: patients over the age of 18, of both sexes, who had been diagnosed with OSCC in stage T1/T2N0M0 and who had undergone the SLNB technique following the CHUS protocol. The minimum follow-up period was six months after the SN technique had been performed. The exclusion criteria were: patients with tumours in advanced stages (T3/T4), patients with oral metastasis from other tumours, patients who had undergone a previous surgical procedure and who were in clinical relapse, patients with a tumour with a different histopathological origin than OSCC, and patients who refused to sign the specific informed consent form.

### 2.4. Marking and Study of the Tumour’s Lymphatic Drainage

The presurgical localization of the SN was performed in all patients following the European Association of Nuclear Medicine (EAMN)’s guidelines for a 2-day protocol study [2]. The day before the surgical procedure, a dose of 3mCi (111 MBq) of Thecnecium-99m-labelled nanocolloids (99mTc-nanocolloid) was administered to each patient, distributed in four injections covering each of the cardinal points around the tumour. Planar images were taken using a GE Millennium VG gamma camera in the first 60 min after the injection (LEHR collimator, energy-peak 140 KeV with 20% window, zoom 1 256 × 256 matrix). Where required, and after the planar images had been taken, a SPECT-TC cervical image was also taken using a GE Optima NM/CT 640 gamma camera. During the surgical procedure, intraoperative lymphatic mapping was performed using a portable gamma-probe for radio guidance (Navigator GPS gamma probe) in order to identify the “hot spots” corresponding to the SN. The primary tumour was often resected first, particularly in those cases in which the primary tumour was close to potential adjacent SLNs in order to avoid the “shine-through effect” that would cause the nearby radioactive SLNs to be occulted by the activity of the primary lesion, therefore limiting the accuracy of the SLN detection. After the SLNs were removed, the surgical field was scanned to ensure that no SLNs were left. SNLs were considered as any node with an uptake higher than 10% of the ex-vivo counts of the hottest node.

All of the extracted lymph nodes were sent to the Pathological Anatomy Service for their histopathological study. If no OSCC metastasis was observed in the histological study of the lymph nodes, the patient did not undergo the neck dissection procedure and they were followed up on a regular basis. If metastasis did appear, the patient was considered to be at an advanced stage.

### 2.5. Histopathological Study of Sentinel Nodes (SN)

Each SN node was formalin-fixed, bisected and each half was routinely processed and paraffin-embedded. For the immunohistochemical study, each block was trimmed by 50 to 100 micrometres, depending on the thickness of the lymph node, and ten three-micrometre tissue sections were taken to special immunohistochemistry-coated slides (Agilent Dako, Santa Clara, CA, USA). The sections were stained as follows: No. 1, 2, 4–7, 9 and 10 with haematoxylin-eosin and 3 and 8 with cytokeratin.

For the immunohistochemistry (IHC) analysis we used Cytokeratin AE1/AE3 (Clone AE1/AE3, FLEX Ready to use, Agilent Dako, Santa Clara, California (CA), USA), with automated equipment (Omnis, Dako, Santa Clara, CA, USA). To summarise, the epitope retrieval was performed in 10 mM sodium citrate buffer (pH 9.0) using a water bath for 40 min at 95–99 °C. Endogenous peroxidase was blocked with a peroxidase-blocking reagent (Agilent Dako, Santa Clara, CA, USA) for 5 min. Incubation with the primary antibody was performed at room temperature for 20 min and the staining was revealed with EnVision (20 min) and DAB (10 min) (Agilent Dako, Santa Clara, CA, USA). Finally, they were counterstained with HE for 15 min.

### 2.6. Information and Variables

Information was gathered at different stages of the treatment: pre-surgical, surgical, post-surgical and follow-up, and, likewise the following clinical, histopathological and follow-up details were recorded: sex, age, location, TNM (Tumor Node Metastasis), T1 or T2, classification, type of cervical surgical treatment, largest size of SN (cm^2^), histopathological involvement of SN (negative, micrometastasis, macrometastasis), clinical relapse and death. The following dates were gathered for the follow-up assessment: birth, surgery, relapse, successes, therefore making it possible to calculate the follow-up time, disease-specific survival (DSS), disease-free survival (DFS). The diagnostic indices were calculated: true positives (TP), true negatives (TN), false positives (FP), false negatives (FN), positive predictive value (PPV), sensitivity, specificity and negative predictive value (NPV).

### 2.7. Statistical Analysis

The variables were expressed as a frequency and a percentage or mean ± standard deviation (SD). 2 × 2 tables were built to determine the diagnostic indices. The Chi-square was used to establish the relationship between mortality and SN. Student’s *t*-test was used to examine the relationship between the DSS and DFS with SN. ROC (Receiver Operating Characteristic) curves were constructed to determine the area under the curve (AUC), the sensitivity and specificity. The Kaplan–Meier curve and the log-rank statistic test were used to estimate survival. Multinomial logistic regression models were constructed to determine the risk of mortality using OR (odds ratio). The statistical analyses were made with IBM SPSS 23 software (IBM Inc., Madrid, Spain). The significance level was established at *p* < 0.05.

## 3. Results

The final sample consisted of 60 patients, 28 (46.7%) men and 32 (53.3%) women, with an average age of 66.9 ± 12.3 years old and a range from 34.9 to 92 years old. 60 tumours were operated on with the SN technique, 32 (53.3%) in stage T1 and 28 (46.7%) in stage T2. Table 1 and Table 2 includes all of the clinical and histopathological data and monitoring of the sample. The average monitoring time was 42.8 ± 23.1 months (from 0.6 to 87.3 months).

With regards to hidden metastasis, 10 patients (16.7%) presented with SN+ (sentinel node positive). The average macroscopic diameter was 1.74 ± 2.68 cm^2^ (from 0.1–17.6 cm^2^). At the histopathological level, the average number of focal points was 1.30 ± (from 1 to 2 maximum) with an average maximum diameter of 4.62 ± 4.62 mm (range from 0.2 to 15 mm). The majority (80%) only presented subcapsular affection; however, one case also presented with extracapsular affection.

With regards to mortality, out of the 50 SN− patients, 39 (78%) survived, 6 (12%) died due to the neoplastic process and 5 (10%) died due to causes unrelated to the tumour. Out of the 10 SN+ patients, 3 (30%) survived, 6 (60%) died from cancer and 1 (10%) died from non tumour-related causes. The survival of SN− patients was higher than SN+ patients in this follow-up period (*p* = 0.002).

The DSS for the SN− patients was 27.2 ± 22.8 months and 21.1 ± 12.5 months for SN+ patients; the DFS was 17.0 ± 15.1 months for SN− patients and 13.6 ± 8.9 months for SN+ patients. Using the Kaplan–Meier curves, we determined that the average survival estimation for SN− patients was 74.0 months (CI95% 67.6–80.5) and it was 45.4 months (CI95% 10.9–24.0) (*p* = 0.002) for SN+ patients (Figure 1). Using the proportional hazards model, we determined that sex played a significant role in the survival time, with women presenting a HR = 5.889 (CI95% 1.01–31.500; *p* = 0.038) (Figure 1B). However, SN does not offer significant differences in the DFS for locoregional recurrence.

By using logistic regression, we determined that SN+ patients presented an OR = 11.00 (CI95% 2.393–50.589, *p* = 0.002) for cancer-related mortality. When we adjusted the regression model to a location different to the tongue and stage T2, the OR for mortality in SN+ patients was 28.75 (CI95% 2.71–305.62, *p* = 0.005) (Table 3).

In terms of the diagnostic performance of the SN test, based on TVP = 10, TVN = 42, TFP = 0 and TFN = 8, 55% sensitivity, 100% specificity, 100% PPV and 84% NPV were obtained. The analysis using ROC curves revealed an AUC = 0.671 (CI95% 0.492–0.850, *p* = 0.046) (Figure 2).

## 4. Discussion

In this study, a greater and significant survival of SN– patients was determined: 78% compared to 30% of the SN+ patients. In quantitative terms, the mean survival estimation for SN– patients was 74.0 months (CI95% 67.6–80.5) and for SN+ patients it was 45.4 months (CI95% 10.9–24.0). For cancer-related mortality, SN+ patients presented an OR of 11.0, and this figure reached 28.75 for T2 cases. The SLNB reached a sensitivity of 55%, however the NPV of 84% is particularly worth mentioning.

SNLB is based on the early detection and correct staging of neck lymph node involvement, since multiple studies have shown that lymph node involvement is decisive for patient survival [3,4,5,6]. In addition, it is not only crucial for survival, but it is also a factor that will determine the patient’s quality of life, when taking into consideration the side effects of the other different treatment options [14].

One of the most relevant figures for the assessment of SLNB is the overall survival, in our study the overall survival of SN+ patients was 40%, which was significantly lower than the overall survival of SN– patients (*p* = 0.002). Kovács et al. [23] also described an overall survival of 38% for SN+ patients and 85% for SN– patients, very similar figures to those obtained in this study. The Sentinel European Node Trial (SENT), a prospective multicentre study (7), put the 3-year overall survival of SN– at 88%. However, some studies have shown fewer differences between groups. For example, Flanch G.B et al.’s [24] German multicentre study with n = 62, showed an exceptional 5-year overall survival for both SN+ patients (79.7%) and SN– patients (92.7%). In their sample of 234 patients, Moya-plana et al. [4] observed that the overall survival of SN– patients at five years was 77.3%, and Cramer et al. [20] observed an overall survival of SN– patients of 82%, figures similar to those obtained in this study. Furthermore, according to the data obtained in more recent studies, there is a higher probability of mortality in SN+ patients [25]. In our study there was also a significant difference in the estimated survival times of patients who obtained SLN+ and SLN–, with these recorded at 45.4 months and 74, respectively.

Furthermore, the sensitivity of a diagnostic test is essential in order to determine which patients are truly ill, and this figure is among the most reported in studies, although the NPV is much more decisive. Specifically, in our study, of the 50 patients diagnosed with SN–, 10 were given a posteriori diagnosis of hidden lymphatic metastasis, obtaining a NPV of 84% and a sensitivity of 55%. This datum is similar to that presented by Ahmed Al-Dam et al. (19), who reached a sensitivity of 50%. However, it is in contrast to other studies, such the retrospective study of 70 patients by E. Martilla et al. [3] who presented a test sensitivity of 80% and a NPV of 97%; likewise, the study by Loree et al. [25] offered very similar results with a sensitivity of 75% and a NPV of 91%. Very similar data was also extracted from the multicentre studies performed by Schilling et al. [7] (n = 415), Vishnoi JR et al. [26] (n = 134) and Den Toom et al. [27] with a sensitivity of 86%, 93% and 97%, respectively. Some recent meta-analyses [28,29] reported up to 95% sensitivity [5,13]. These results highlights the sentinel node biopsy as a reliable diagnostic staging technique for the clinically negative neck patients with early-stage oral squamous cell carcinoma. However, it is evident that there is a high discrepancy in the different values reported in the literature with these ranging from 55% to 100% [30].

There are a number of factors that determine the variability in the diagnostic performance values of the SLNB: work centre, type of tumour sample (HNC or only OSCC from the oral cavity), follow-up period, date of publication of the studies and the performance of serial SN cuts with or without IHC. According to Liu et al. [29] the subgroup analysis based on IHC indicated that H and E staining combined with IHC was significantly more sensitive that the results obtained when H&E staining was performed on its own, with a sensitivity of 0.88 (CI 95%: 0.86 to 0.90) vs. 0.77 (CI 95%: 0.68 to 0.85). Furthermore, the early publication subgroup (2000 a 2008) had a better combined sensitivity than the late publication subgroup (2009 to 2016) (0.92 [0.87–0.95] vs. 0.86 [0.83–0.88].

According to Liao et al.’s [31] recent meta-analysis, which contained 73 articles on HNCs, it appears that, despite its limitations, when compared with other techniques SLNB presents the best results for detecting hidden metastases. The sensitivity for fine needle aspiration (FNA) puncture was 56.4%, it was 84.9% for SLNB, 47% for computed tomography (CT), 56.6% for MRI, 48.3% for PET and 63.3% for ultrasound.

This work contributes to improving knowledge in the field of SLNB of tumours of the oral cavity, supporting the European Association for Craniomaxillofacial Surgery (EACMFS) guidelines for the management of clinically N0 tumours, The strength of this study is that it boasts a follow-up period of longer than 3 years and a histopathological analysis technique combined with IHC. The limitations of this study include its modest sample size, given that it was a single-centre study and the inherent limitations of possible diagnostic bias.

## 5. Conclusions

Patient mortality in T1/T2-N0 remains very high. The use of the SLNB in these patients seems to have an adequate sensitivity for the detection of hidden metastases. The histopathological study by immunohistochemical analysis seems to be key for the determination of micrometastasis and in the improvement of the diagnostic cost-effectiveness of the test. A definitive protocolisation of the SLNB technique in OSCC seems to be necessary.

## Figures and Tables

**Figure 1 medicina-57-01092-f001:**
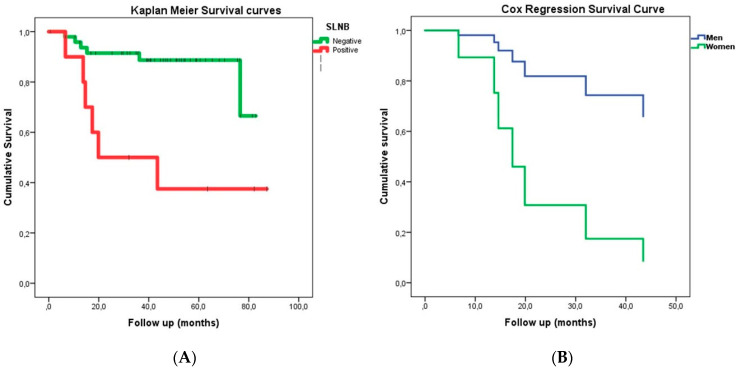
Kaplan–Meier (**A**) and Cox (**B**) survival curves. (SLNB: Sentinel lymph node biopsy technique).

**Figure 2 medicina-57-01092-f002:**
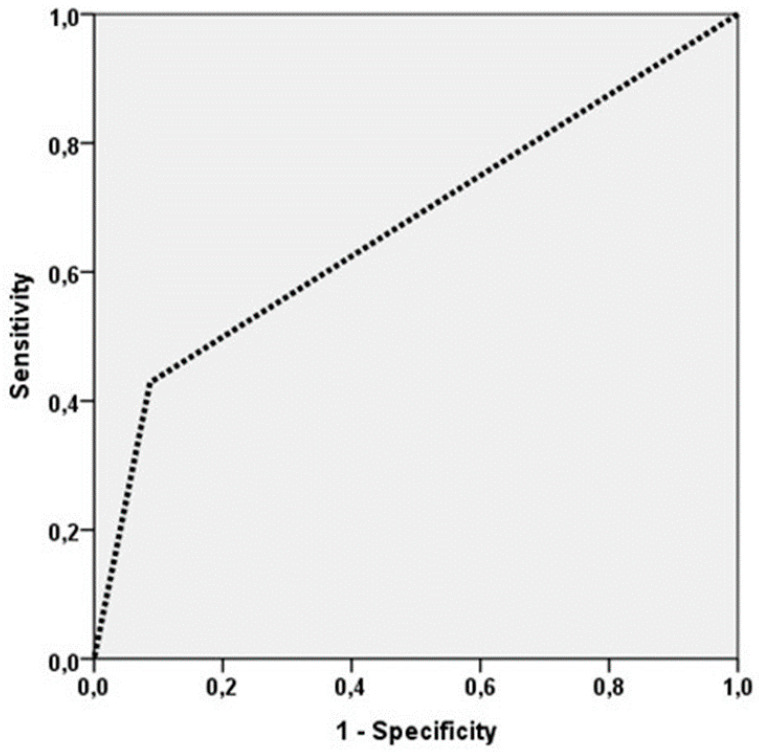
ROC curve for the sentinel lymph node biopsy (SLNB) technique, representing the sensitivity and 1-Specificity.

**Table 1 medicina-57-01092-t001:** Descriptive summary of the qualitative variables of the patients included in the study.

Variables		Number	%
Gender	Male	28	46.7
	Female	32	53.3
Location	Tongue	24	40.0
	Floor of mouth	13	21.7
	Buccal mucosa	8	13.3
	Retromolar trigone	3	5.0
	Gum	4	6.7
	Hard palate	1	1.7
	Soft palate	1	1.7
	Alveolar ridge	5	8.3
	Upper lip	1	1.7
TNM(Tumor, Nodes, Metastases)	T1	32	53.3
	T2	28	46.7
SN (Sentinel Node)	Negative	50	83.3
	Positive	10	16.7
Neck Metastasis Post-SN	No	46	76.7
	Yes	14	23.3
Histopathological Lymph Node Description	Negative	50	83.3
	Micrometastasis	2	3.3
	Macrometastasis	8	13.3
Histopathological Lymph Node Affectation	Subcapsular	8	(80)
	Intraparenchymal	1	(10)
	Complete (both)	1	(10)
Extracapsular Affectation	No	9	(90)
	Yes	1	(10)
Neck Treatment	No treatment	50	83.3
	Functional dissection	10	16.7
Development	Stable	32	53.3
	Relapse	17	28.3
	Recurring Relapse	1	1.7
	Death	10	16.7
Locoregional Recurrence	No	38	63.3
	Yes	22	36.7
Death	No	42	70.0
	Yes	12	20.0
	Yes, for other reasons	6	10.0

**Table 2 medicina-57-01092-t002:** Descriptive summary of the quantitative variables of the patients included in the study.

	Average (SD)	Minimum-Maximum
Age of Diagnosis (Years)	66.9 (12.3)	34.9–92.0
SN Size (cm^2^)	1.7 (2.7)	0.1–17.6
Follow-up Time (months) SN− (sentinel node negative) SN+ (sentinel node positive)	42.8 (23.1) 43.8 (21.8) 38.1 (29.7)	0.6–87.3 0.6–82.8 6.7–87.3
Specific Survival (months) SN− SN+	24.8 (19.3) 27.2 (22.8) 21.1 (12.5)	0.6–76.6 0.6–82.8 6.7–43.5
Disease-Free Survival (months) SN− SN+	15.9 (13.3) 17.0 (15.1) 13.6 (8.9)	1.8–61.3 3.9–61-3 1.8–29.9

**Table 3 medicina-57-01092-t003:** A univariate logistic regression analysis was performed to determine the univariate OR for death. The statistical analysis of the adjusted OR was performed using gradual multivariate logistic regression adjusted for SLN, location and TNM. SLN: sentinel lymph node; OR: Odds Ratio.

Covariate	Death			
	Univariate OR (95% CI)	*p* Value	Adjusted OR (95% CI)	*p* Value
SLN (Sentinel Lymph Node)				
Positive vs. Negative	11.00 (2.39–50.59)	0.002	28.75 (2.71–305.62)	0.005
Location				
Other vs. Tongue	4.23 (0.84–21.40)	0.081	18.61 (1.34–258.65)	0.029
TNM				
T2 vs. T1	4.58 (1.10–19.11)	0.037	5.15 (0.86–30.72)	0.072

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
