# Peer review of "Analysis of the Efficiency and Prognostic Value of the Sentinel Node Technique in Oral Squamous Cell Carcinoma after Seven Years"

_medicina, 2021, doi:10.3390/medicina57101092_

Round 1

Reviewer 1 Report

Very interesting and useful article.

Very well written 

Please verify the word "ganglions", and possibly replace it

Otherwise accept and publish it

Author Response

Reviewer #1

Comments and Suggestions for Authors

Very interesting and useful article.

Very well written

Please verify the word "ganglions", and possibly replace it

Otherwise accept and publish it

Thank you very much for the positive comments. The wrong word has changed along with the manuscript.

Reviewer 2 Report

In this manuscript Ajuria MS and colleagues has demonstrated that use of the sentinel lymph node biopsy technique in oral squamous cell carcinoma patients offers adequate sensitivity for the detection of hidden metastases and help in improving patient’s survival with negative test. I read this article with great interest and appreciate the author’s hard work in long term follow-up.

I have major concern regarding the novelty of the study. Earlier studies ( many of them are cited by the authors) have already shown that the overall survival is significantly higher in SN negative patients compared to SN positive patients. Also the conclusion demonstrating Sentinel node biopsy as a reliable diagnostic staging technique for the clinically negative patients with early-stage (T1–T2, N0) oral squamous cell carcinoma is well discussed. Unfortunately, I do not see much benefits for the readers from this manuscript.

Minor concern

  1. The authors should check the typographical errors (eg. Line 77 and 181) throughout the manuscript.
  2. The authors should split Table 1 in two separate tables for the ease of the readers.

Author Response

Comments and Suggestions for Authors

In this manuscript Ajuria MS and colleagues has demonstrated that use of the sentinel lymph node biopsy technique in oral squamous cell carcinoma patients offers adequate sensitivity for the detection of hidden metastases and help in improving patient’s survival with negative test. I read this article with great interest and appreciate the author’s hard work in long term follow-up.

We thank the reviewer for his/her positive comments.

I have major concern regarding the novelty of the study. Earlier studies (many of them are cited by the authors) have already shown that the overall survival is significantly higher in SN negative patients compared to SN positive patients. Also the conclusion demonstrating Sentinel node biopsy as a reliable diagnostic staging technique for the clinically negative patients with early-stage (T1–T2, N0) oral squamous cell carcinoma is well discussed. Unfortunately, I do not see much benefits for the readers from this manuscript.

We appreciate the comments, and we would like to briefly highlight the importance and the strengths of our manuscript.

In the cases of breast cancer and melanoma, the SN technique is highly implemented and protocolized, however, this is not the case for OSCC patients. For years, the management of T1-T2 stage OSCC patients has been the subject of much debate, with unilateral neck dissection, a watch and see policy, and, finally, the SN technique having been considered. No consensus has been reached in this regard; however, the latter appears to be the best and most promising alternative. As a result, several guidelines/protocols and recommendations for this technique have already been published, all of which coincide that a multidisciplinary team (nuclear medicine, radiology, maxillofacial surgery, and pathological anatomy) is an essential requirement. It is worth mentioning that only recently in 2018, in the Eighth International Symposium on the Sentinel Node Technique, several experts in this field gathered together to give a number of scientific-evidence-based recommendations focusing on the surgical aspects. The 17 recommendations ranged from the pre-operative, and intra-operative aspects of the patient, to the new technologies that can be applied with this technique. Hence, probably not all the studies and protocols used before that consensus have been based on these rules, and this fact makes necessary new clinical trials taking into account these specific recommendations.

Moreover, our manuscript comprises other pathological aspects, such as the number of focal points and diameter of the micro-metastasis, everything related to the use of immunohistochemistry in frozen sections, which have been recently related with differences in diagnostic capacity at that microscopic level. All of them joined to a quite far average follow-up which may help to consolidate this technique in the head and neck oncology area.

Minor concern

The authors should check the typographical errors (eg. Line 77 and 181) throughout the manuscript.

Thank you very for the deep analysis of our manuscript which helps to improve significantly the quality of the given information. The manuscript has been double-checked again by the native team and the Editorial will also do it in case of acceptance.

The authors should split Table 1 in two separate tables for the ease of the readers.

We agree with the reviewer and we have performed to different tables.

Reviewer 3 Report

The authors present a paper about "Analysis of the efficiency and prognostic value of the sentinel node technique in oral squamous cell carcinoma after seven years".

The topic is of great interest and has been addressed by several authors in the recent years.

The methodology seems to be adequate and the prospective design of the study population are points of strength.

I have however some major concerns which I would like the authors to address

1) How were the patients staged? (TC, MRI, PET-CT)?

2) What was the subsequent therapeutic strategy for SN+ patients (RT, Chemo)?

3) 60% of patients dead due to cancer in T1/T2 SN+ is an high percentage of death: how do the authors explain it? which were the adjuvant therapies provided (RT? Chemo?)

4) The final statement of the authors is "The use of the SLNB in these patients seems to have an adequate sensitivity for the detection of hidden metastases, improving survival in patients with negative test". Such statement is difficult to undertand beacause those patients with SN- who have had the same prognosis also without SLNB. So which is the proposed role of SLNB: prognostic (if so it should be clearly stated)? predictive (if so authors should explain the 60% of SN+ patients dead.

Author Response

Comments and Suggestions for Authors

The authors present a paper about "Analysis of the efficiency and prognostic value of the sentinel node technique in oral squamous cell carcinoma after seven years".

The topic is of great interest and has been addressed by several authors in the recent years.

The methodology seems to be adequate and the prospective design of the study population are points of strength.

I have however some major concerns which I would like the authors to address

1) How were the patients staged? (TC, MRI, PET-CT)?

Following the International guidelines and protocols, all the patients were staged by computed tomography (CT) of the head and neck area. Once staged, those that do not present lymph nodes suspicious of radiological metastasis in the CT scan are proposed for SLNB with the protocol described. The evaluation of these lymph nodes includes: a) Size, b) Morphology, c) Loss of the fatty hilum, d) Necrosis, e) Contours and homogeneity, g) Calcifications, h) The grouping of 3 or more lymph nodes in the same ganglionic territory, each with a diameter of between 8 and 15 mm suggests that the nodes are pathological, i) Vascular involvement.

2) What was the subsequent therapeutic strategy for SN+ patients (RT, Chemo (CMT))?

Positive patients in the sentinel node technique are treated as advanced stages with RT and CMT and complete neck dissection in the next 24-48 hours.

3) 60% of patients dead due to cancer in T1/T2 SN+ is an high percentage of death: how do the authors explain it? which were the adjuvant therapies provided (RT? Chemo?)

This is because the most important prognostic factor in HNSCC is the presence of positive lymphadenopathies, N+ (from TNM) is an independent prognostic factor of DFS, OS, and DSS, therefore even with adjuvant therapies (elective neck dissection, CMT, and RT) the only presence of an N+, conditions the prognosis and mortality of patients.

4) The final statement of the authors is "The use of the SLNB in these patients seems to have an adequate sensitivity for the detection of hidden metastases, improving survival in patients with negative test". Such statement is difficult to undertand beacause those patients with SN- who have had the same prognosis also without SLNB. So which is the proposed role of SLNB: prognostic (if so it should be clearly stated)? predictive (if so authors should explain the 60% of SN+ patients dead.

We apologize for this misunderstanding. In fact, the prognosis is not improved by the surgical technique but by the condition of being an initial stage. The conclusions section has been accordingly modified.

Round 2

Reviewer 2 Report

Thank you for your revised manuscript. I understood that the authors tried to do their best. I would recommend the authors to cite the article PMID: 24677355 highlighting Sentinel node biopsy a reliable diagnostic staging technique for the clinically negative neck in patients with early-stage oral squamous cell carcinoma.

Author Response

Reviewers' comments:

Reviewer #2

Comments and Suggestions for Authors

“Thank you for your revised manuscript. I understood that the authors tried to do their best. I would recommend the authors to cite the article PMID: 24677355 highlighting Sentinel node biopsy a reliable diagnostic staging technique for the clinically negative neck in patients with early-stage oral squamous cell carcinoma”.

Thank you very much for your comment, it was very useful to our article. We have added the cite on page 9, lines 261-265 and lines 374-375.

Reviewer 3 Report

I have no further comments

Author Response

Thank you very much for your comments